# Development of Novel Method for Immobilizing TMAH-Degrading Microbe into Pellet and Characterization Tool, for Verifying Its Robustness in Electronics Wastewater Treatment

**DOI:** 10.3390/ijerph17124411

**Published:** 2020-06-19

**Authors:** Seungjoon Chung, Jeongyun Choi, Jinwook Chung

**Affiliations:** R&D Center, Samsung Engineering Co., Ltd., Suwon 16523, Korea; phd.chung@samsung.com

**Keywords:** differential scanning calorimetry (DSC), electronics wastewater, immobilized cells, polymerization, tetramethyl ammonium hydroxide (TMAH)

## Abstract

This study describes an immobilization method of enriched microorganism, for robustly degrading organic compounds, including tetramethyl ammonium hydroxide (TMAH) in electronics wastewater without an increase of total organic carbon (TOC) in effluent. The enriched TMAH degrading bacteria was entrapped inside the pellets through polymerization. Polymerization conditions were optimized in terms of long-term TOC leak tests of pellet. Among several methods, a differential scanning calorimetry (DSC) analysis was found to be effective for the hands-on evaluation of stability in pellet. Stable pellets showed less than 10 J/g of curing heat by DSC analysis. This method is suitable for the optimization of polymerization conditions and controlling the quality of pellets. The removal efficiency of TMAH was over 95% and effluent concentration of TOC was below 100 ppb. The viability test results revealed that entrapped microorganisms were actively survived after five months of operations. This immobilization method is strongly suggested as a new strategy for the wastewater reuse process in low-strength electronics wastewater.

## 1. Introduction

Semiconductor and display manufacturer in the electronics industry consume large quantities of process water with extremely high purity (ultrapure water, UPW) and produce as much wastewater with complex composition. On the one hand, the limited surface water supply in Korea caused a significant water scarcity issue demanding wastewater reuse. On the other hand, the complex composition of persistent organic compounds and higher operating expense limit application of wastewater reuse in electronics industry [1].

High-strength wastewater contains various kinds of etchants and organics solvents with high concentrations. Low-strength electronics wastewater is produced by rinsing of residual etchant and organic solvent such as tetramethyl ammonium hydroxide (TMAH) and isopropyl alcohol (IPA) with UPW. Thus, the composition of this wastewater is relatively less complex and, consequently, the treatment process could be simpler than high-strength wastewater. Therefore, low-strength wastewater treatment for reuse is getting more attention and electronics manufacturers tend to separate low-strength wastewater from high-strength wastewater [2,3].

Biological treatment would be a suitable choice for the treatment of low-strength electronics wastewater, because carbonaceous contaminants are converted into less toxic and less mobile species without much operating expense [4,5]. Therefore, many research studies were performed on the removal of TMAH, which is a dominant species in electronics wastewater. Only a few research studies were performed in low-strength wastewater, while most of them were in high-strength wastewater. The biological removal of TMAH in low strength electronics wastewater is challenging, because the toxicity of this chemical rendering removal of total organic carbon (TOC) to ppb levels, by existing biological treatment method such as activated sludge process [1].

As an alternative to activated sludge, biological activated carbon (BAC) is an option for reuse of low-strength electronics wastewater. It provides many benefits such as small footprint, inexpensiveness and simplicity of approach, compared with activated sludge. However, this process often causes TOC contamination and subsequent fouling of membrane process such as ultrafiltration in UPW production facility, because of detached microorganisms from activated carbon. Thus, there is a strong need for a biological treatment technology without bleeding of the microorganism.

To overcome this limitation, the immobilization of microorganisms by polymerization was considered in this study, for enhancing the rate of biodegradation and minimizing inhibition by toxic chemicals in wastewater. In the literature, several polymerization techniques for immobilizing microorganism were reported, such as immobilization by acrylamide [6], polyvinyl alcohol [7,8], alginate [9], and polyethylene glycol (PEG) [10,11]. Among the immobilization method, it was reported that the PEG-based method had good mechanical strength [12,13]. Most of these research studies were focused on the application in nitrification and various organic compounds in municipal wastewater [14,15]. In low strength electronics wastewater, contamination of the TOC from microorganism and its support would be comparable to the concentration of TOC in effluent from wastewater treatment facility for reuse. However, no research studies have been performed on the stability of prepared polymer, despite its importance in low-strength wastewater.

Previous research studies in our laboratory revealed that the polymerization with PEG based prepolymer is effective for the immobilization of TMAH degrading bacteria, and suggested the application of the immobilized microorganism in the treatment of low-strength electronics wastewater [11]. Moreover, the enrichment method of TMAH degrading bacteria and fundamental microbial kinetics was investigated [16]. However, long-term stability of pellets was not fully investigated, which has been rarely studied elsewhere in the literature. In this study, we focused on the development of an immobilization method for maintaining high removal efficiency without the release of TOC during operation. We evaluated the performance of a biological process using stable highly cross-linked pellets immobilized TMAH degrading bacteria for the treatment of low-strength electronics wastewater.

## 2. Materials and Methods

### 2.1. Enrichment of TMAH Degrading Bacteria

The activated sludge from a wastewater treatment plant (Kiheung, Korea) was mixed with 5 g/L of TMAH and 300 mL of mineral solution (1000 mg/L of (NH_4_)_2_SO_4_, 700 mg/L K_2_HPO_4_, 300 mg/L of KH_2_PO_4_, 500 mg/L of MgSO_4_, 0.6 mg/L of H_6_BO_6_, 0.4 mg/L of CoSO_4_·7H_2_O, 0.32 mg/L of ZnSO_4_·7H_2_O, 0.082 mg/L of MnCl_2_·4H_2_O, 0.0648 mg/L of Na_2_MoO_4_·2H_2_O, 0.064 mg/L of NiCl_2_·6H_2_O, 0.028 mg/L of CuSO_4_·5H_2_O, 0.5 mg/L of FeSO_4_·7H_2_O, and 200 mg/L of CaCl_2_.2H_2_O). The mixture was kept in a dark place at 30 °C for the enrichment of TMAH-degrading bacteria. After concentration of total organic carbon (TOC) in the mixture reached 200 mg/L, the mixture was concentrated by centrifuge. The concentrate was amended by adding an equivalent volume of fresh mineral solution for the further enrichment and the concentration by centrifugation. This fed-batch cycle was repeated until enough amount of concentrated culture was obtained. Two sizes of fermenter (2.5 L and 200 L) were operated in the following condition; temperature of 30 °C, DO concentration >3.0 mg/L, air flow rate of 0.5–1.0 sccm, pH 6.9 ± 0.1, agitation speed under 300–800 rpm. After the completion of enrichment, the contents of TMAH-degrading bacteria increased (almost negligible at the beginning) up to 1.5%.

### 2.2. Polymerization Methods

The enriched TMAH-degrading bacterium (*Mycobacterium* sp. TMAH-W0418) and activated sludge were suspended in a polyethylene glycol (PEG) aqueous solution, containing 22% (w/v) of PEG prepolymer (oligomer 2910, Miwon, Korea), 0–2.4% (w/v) of crosslinker (diallylamine or triallylamine), 0.25% (w/v) of initiator (potassium persulphate), and 1.05% (w/v) of additive (acetic acid), as described in the author’s previous work [17]. After the agitation of the mixture for 1 min, 0.5% (w/v) of promoter (N,N,N’,N’-tetra-methyl ethylene diamine) was spiked and mixing was followed for 5 sec. Then the mixture was introduced to a plastic tube (polyvinyl chloride, inner diameter of 4 mm) and was left for 10 min at room temperature for solidification. Thus, the elastic gel containing enriched TMAH-degrading bacteria with activated sludge was formed inside the tube. The pellets were obtained after it was extruded from the tube, cut at a length of 4 mm, and washed thoroughly with deionized water.

### 2.3. Characterization of Pellets without Microorganism

The structure of the prepolymer was analyzed by a nuclear magnetic resonance (NMR) spectroscopy (AMX 500, Bruker, USA), operating at 75 MHz with a standard carbon 5mm probe, in the presence of the static magnetic field of 7T. The measurements were done in CDCl_3_ at room temperatures (20 to 24 °C). Moreover, ^1^H NMR and ^13^C NMR spectra were recorded to assign peaks of prepolymer. The solid state NMR spectra (^13^C CP/MAS) of prepared pellet were measured by another NMR spectroscopy (DSX 300 MHz, Bruker, Billerica, MA, USA) at the same magnetic field and the frequency of liquid state NMR. The pellets were cut and placed into 4 mm ZrO_2_ tube. The spinning rate varied 4–15 kHz and the temperatures were kept ambient.

The prepared pellet was characterized by pyrolysis–gas chromatography (Py-GC) following the procedure in the literature [18]. A gas chromatography (GC-15A, Shimadzu) equipped with vertical type pyrolyzer (PY-2020D, Frontier Lab) was used for the characterization of prepared pellet to obtain chromatogram (Py-GC). A solution of TMAH (25 wt.% in methanol) (Nacalai Tesque, Japan) was used as a reagent of reactive pyrolysis. Then, 100 µg of the sample was placed in the platinum holder and 4 µL of the TMAH solution was added. The sample in the pyrolyzer was heated to 400 °C. The temperature of reactive pyrolysis and the amount of TMAH addition were empirically optimized, for the highest yield of methyl derivatives, which minimized the contribution of thermal cleavages in the polymer chain. A nitrogen carrier gas was introduced with a flow rate of 50 mL/min, for sweeping the pyrolysis products rapidly from the pyrolyzer to the separation column. The flow rate of carrier gas was reduced to 0.8 mL/min at the inlet of the capillary column with a splitter. The temperature of the column (a capillary column, PTE5, 30 m × 0.25 mm, Supelco) was set at 40 °C for 5 min and increased up to 350 °C at a ramping rate of 10 °C/min, for the separation of pyrolyzates with high boiling points. The temperatures of injector and detector (flame ionization detector, FID) were both set to 320 °C. The peaks in the pyrograms were identified by the Py-GC/MS (Shimadzu QP-5050A). Isobutane was used for electron impact ionization and chemical impact ionization.

The thermal characteristics of pellet were analyzed by the differential scanning calorimetry (DSC). DSC (Perkin–Elmer, DSC7) was equipped with a liquid nitrogen-cooling accessory and calibrated with standard of elemental indium. The sample of 10–20 mg was placed in a sealable aluminum dish with a lid. A drop of the solvent was added to maintain excess solvent in sample. The sample was first cooled below the freezing temperature of the pure liquid to −40 °C and then heated up to 300 °C, with a ramping rate of 10 °C/min.

The stability of pellets without microorganisms were evaluated by measuring TOC extracted from prepared pellets in a continuous mode. Pellets were filled in a glass column (4.0 cm ID × 20 cm height, 250 mL) and distilled water was supplied with 8.0 mL/min (HRT = 30 min) in upflow mode. Aeration was implemented for the fluidization of the pellet and the maintenance of dissolved oxygen (DO) level at 3 mg/L in the reactor. The eluent was sampled on a daily basis and the concentration of TOC extracted from pellets was measured with TOC analyzer (Seivers 900, GE, Boulder, USA), equipped with a flame ionization detector.

### 2.4. Evaluation of Microorganism Immobilized Pellets

Wastewater was fed to lab column (same as used for leak test) with a flow rate of 2.0 mL/min (hydraulic retention time, HRT = 2 h) and was allowed for the biodegradation of contaminants for 4 weeks. The effects of pellet inoculation methods on the removal efficiency of TOC and TMAH were evaluated. The biomass mixing ratio of *Mycobacterium* sp. to sludge were from 4:1 to 1:4 (w/w), and total biomass concentration were from 1000 to 7000 mg/L.

With optimized pellets, different operating conditions of the bioreactor were evaluated in terms of removal efficiencies of TOC and TMAH. The HRT in the column varied from 20–40 min, packing density of pellets were 50 to 80% and the recirculation ratio was 0.8 to 2.3. The concentration of TMAH was measured by ion chromatography (Dionex, ICS-3000), coupled with a conductivity detector and a column (4 × 250mm, Ionpac CS17). A mobile phase (CH_3_SO_3_H solution) was supplied at 1 mL/min for 15 min. The concentration in mobile phase was gradually increased from 6 to 24 mmol/L.

The pellet, attached to a sample plate that was immersed in a water container, was cut into 15-μm slices on a vibrating microtome (HM 650V, Microm, Walldrof, Germany). A slice was placed on a slide glass, and 100 µL LIVE/DEAD BacLight viability stain (Molecular Probes), prepared according to the manufacturer’s instructions, was placed atop the slice, and a coverslip was placed over the entire slice. The sample was kept in a dark place for 15 min, and the stains were visualized under an epifluorescence microscope (BX51, BX2 series, Olympus, Tokyo, Japan). Images were captured by digital microscopy (DP71, Olympus, Tokyo, Japan) and processed using analySIS Five LS Research (Olympus Soft Imaging Solutions, Münster, Germany).

## 3. Results and Discussions

### 3.1. Enrichment of TMAH Degrading Bacteria

Figure 1a shows the biomass concentration, pH, TMAH and NH_4_^+^-N concentration over time in the fed-batch culture with six times feeding. To compensate the great oxygen consumption by a cell growth rate, the agitation speed during incubation was adjusted from 500 rpm to 800 rpm. The pH decreased by the consumption of TMAH in the culture medium during the incubation. In the fed-batch culture process, mineral solution was added to reach 5% of the total volume for the adjustment of the TMAH concentration to 10 ~ 30 g/L. After six cycles, the cell density of the culture reached 106 in OD_600_. For a scale-up of the fermenter (200 L), the total incubation time was prolonged by increasing the frequency of feeding, to obtain higher cell density in the large fermenter. The concentration of NH_4_^+^-N was kept below 500 mg/L against the inhibition of free ammonia. After feeding nine times and culturing at 130 h, the cell density of the fermenter reached 182 in OD_600_ (see Figure 1b).

### 3.2. Preparation Conditions of Pellets

The one of the key objective in this study is the preparation of a stable polymeric support under low-strength electronics wastewater. The wastewater was mainly composed of several organic solvent such as TMAH and IPA, which extract TOC from polymeric media. In addition, wash out by shear of the pellets during fluidization, and diffusion by concentration difference would increase TOC level in the effluent. TOC extraction is generally not a concern at high-strength electronics wastewater. However, the TOC extraction from pellets to treated water in sub-ppm level would cause a significant reduction of removal efficiency at low-strength wastewater. Therefore, a highly cross-linked polymer is required to avoid TOC extraction. The preliminary TOC extraction tests were performed under a different type of prepolymer, concentration of crosslinker and concentrations of initiator (not shown). The partial list of preliminary test results was enlisted in Table 1. PEG monomer was not able to provide sufficient crosslinks in the polymer, leading to TOC breakthrough in the long term test. As a result, PEG monomer was not successful, irrespective of preparation conditions, which implied that more crosslink in the polymer is necessary. Most importantly, a PEG oligomer with more than 1.6% of crosslinker was effective for the preparation of a stable pellet (see Table 1). Figure 2 shows representative TOC extraction test results, with selected crosslinker concentrations for comparison (1.0% and 1.6%). Effluent from pellets prepared 1.0% of crosslinker, which shows a breakthrough of TOC after 70 days, while those with 1.6% do not even after a year. It is reasonable that this difference was caused by unreacted functional groups in the prepolymer, due to an insufficient amount of crosslinker. This instability of pellet is not easily observed in the literature, because researchers commonly perform short-term tests (1–2 months).

### 3.3. Characterization of Pellets

The objectives in this section are to find the reason of instability in pellets and the method of characterization for the quality control of pellets. First, the characteristics of prepolymer were investigated. The chemical structure of PEG oligomer was analyzed by NMR spectra. ^13^C and ^1^H-NMR spectra were matched with the structure of the prepolymer, as shown in Figure 3a–c. Analysis results indicated that the oligomer is composed of PO/EO copolymer blocks, urethane blocks, and acrylate blocks. Figure 4 shows solid state NMR analysis results of pellets prepared with and without crosslinker. The peak of acrylate block was almost negligible if the concentration of crosslinker was sufficient (higher than 1.6%). Meanwhile, the broad peak of acrylate was identified in the sample without crosslinker, which indicates that the presence of acrylate block in the pellets would be the reason for instability causing the TOC leak. Figure 5 shows the reactive pyrolysis GC-MS results of oligomer solution and extracts from prepared pellet with 1.0% of crosslinker which exhibits a TOC leak. The spectrum of acrylate was detected on both of the samples. Thus, it is analogous that the unreacted acrylate block is a cause of TOC leak.

To quantify the degree of polymerization, thermal characteristics of prepared pellets were analyzed by DSC, as shown in Figure 6a. The DSC thermograms of (I)–(V) correspond to cases of 0–2.4% of crosslinker addition. The area of curing peak (heat of curing) tends to decrease as the concentration of crosslinker increases. In the literature, an exothermic curing peak of methyl methacrylate typically occurs at about 100 °C [19]. It is postulated that the degree of polymerization in pellets has a relationship with the heat of curing in methyl methacrylate. Higher crosslinker concentrations used for pellet preparation showed lower areas of curing peaks from prepared pellets. In Figure 6b, the areas of curing peak (heat of curing) were presented, relative to the amount of crosslinker. The heats of curing were less than 10 J/g in pellets prepared with higher than 1.6% of crosslinker (without unreacted acrylate), while those were 20~50 J/g less than 1.0% (with unreacted acrylate). The TOC leak test results in Figure 2 confirmed this relationship between the stability of pellets and heat of curing. This means that the degree of polymerization in the prepared pellets can be quantitatively determined by the area of curing peak from DSC analysis results, which is important because this method can be used for optimization of preparation conditions. Other methods, such as solid NMR and reactive pyrolysis GC-MS, were not suitable for the quantification of the stability of prepared pellets. According to the AVOVA test as statistical analysis, the relationship of pellet stability (TOC leakage in effluent less than 50 μg/L over 1 yr) to DSC analysis result (area of curing curve <10 J/g) was statistically significant (*p* < 0.05), while that to other analysis results were not (*p* > 0.05). Therefore, we determined this method as a characterization method for quality control of prepared pellets. This method is a time-saving method compared with TOC leaking test of 1 year. Instead of the chemical-based curing method, the UV-based curing method was considered for continuous production of microorganism immobilized pellet. The UV-based method is advantageous in the continuous production facility over the chemical-based method, due to short gelation time, uniform absorbance in viscous flow, and reaction in the mild condition [20]. Thus, a modified procedure by the UV-initiated polymerization of pellets was developed in the following. UV-initiated polymerization was performed with a bench scale UV curing system (LH-6, Fusion UV systems, Gaithersburg, MD, USA). A UV filter for 300~400 mm in wavelength was installed in front of the UV lamp to avoid the sterilization of the enriched microorganism. The aqueous PEG solution contained 0.02 % (w/v) of 2-hydroxyt-2-methylpropiophenone as a photoinitiator, instead of the chemical initiator and the promoter. The same ratio of PEG prepolymer, crosslinker, and concentrated culture were injected in the aqueous solution. The mixture was placed on the top of the quartz plate with a molding template of 4 cm x 4 cm with a 2 mm in height. The UV was exposed to the mixture at both ends (top and bottom) of the template, with different exposure times for the initiation of polymerization. The UV-polymerized pellets were obtained by cutting hydrogel into 4 mm × 4 mm. Figure 6c shows the area of the curing curve from DSC analysis, relative to the duration of curing by UV. The heat of curing was below 10 J/g when UV was exposed to pellet for longer than 3 min. The TOC leak test was followed, to verify the stability of prepared pellet as shown in Figure 7. The test results agreed with the previous results with the chemical initiator. Thus, the UV exposure time for the preparation of a stable pellet was fixed to 3 min.

### 3.4. Characteristics of Wastewater

The wastewater was obtained from a local semiconductor manufacturer (Kiheung, Korea). The characteristics of wastewater were summarized in Table 2. The concentration of TOC was about 4.4 mg TOC/L. The wastewater includes approximately 2 mg TOC/L of TMAH, 1.6 mg TOC/L of IPA, 0.6 mg TOC/L of methanol, and a trace amount of ethanol and acetone. The summation of these compounds account for more than 96% of TOC. As mentioned earlier, this low-strength electronics wastewater has a simple composition. The total nitrogen was mostly in the form of organic nitrogen (TMAH), which can be transformed to ammonia nitrogen during biodegradation.

### 3.5. Optimization of Inoculation Method and Operating Conditions

Figure 8a,b show the concentrations of TOC and TMAH in the effluent of the reactor packed with pellets in different biomass mixing ratios (ratios of *Mycobaterium* sp. to sludge) and different total biomass concentrations. In the biomass mixing ratio, 2:3 (w/w) of *Mycobaterium* sp. to sludge appeared efficient, because TMAH concentration increased above 0.5 mg/L at the 3:2 and 1:4 mixing ratio. As shown in Figure 8b, TMAH and TOC concentrations tend to decrease as the total biomass concentration inside the immobilized pellets increased. The lowest effluent concentrations were obtained at a biomass concentration of 7000 mg/L. Therefore, a biomass mixing ratio of 2:3 (w/w) and a total biomass concentration of 7000 mg/L were chosen for the inoculation of the pellet.

Operating conditions of the pellet reactor were evaluated under various HRTs, packing densities, and recirculation ratios at DO concentrations of 1.5 mg/L or higher. TOC and TMAH removals were examined in different HRTs (20, 25, 30, and 40 min), with a packing density of 70% and a recirculation ratio of 1.8 Q. As shown in Figure 9a, there was little difference between TOC and TMAH removal efficiencies at HRTs of 30 and 40 min, but removal efficiencies gradually decreased from the HRT of 20 min. The lower removals of TMAH resulted from a lower contacting time between pollutants and microorganisms.

To compare the removal of TOC and TMAH by packing density, the HRT was set to 30 min and the recirculation ratio was maintained at 1.8 Q. The packing density of pellets in the reactor varied between 50%, 60%, 70%, and 80%. As shown in Figure 9b, TMAH removal efficiencies have seldom changed regardless of packing density, while the average removal efficiencies of TOC were 52%, 61%, 63%, and 63% at packing ratios of 50%, 60%, 70%, and 80%, respectively. This result implies that over 60% of packing density is required to maintain maximum TOC removal.

The system was operated for 60 d or more, with varying recirculation ratios at a packing density of 70% and HRT of 30 min. The DO concentration in the reactor was held through intermittent aeration of the treatment tank. As shown in Figure 9c, the average TOC concentration at 2.0 Q of recirculation ratio was 932 μg/L, which was the highest removal efficiency. At higher recirculation efficiencies than 2.0 Q, the linear velocity in the reactor increased with the lower opportunity for microorganisms, to make contact with pollutants leading to decreasing the efficiency. The linear velocity and recirculation ratio were 24 m/h and 2.0 Q, at the highest efficiencies of TOC and TMAH removal. This result indicates that recirculation ratio is the dominant factor for removal efficiency among three operational factors.

### 3.6. Viability of Inoculated Pellets

The LIVE/DEAD BacLight viability kit (Molecular Probes Inc.) was used to differentiate living and dead bacteria, based on the plasma membrane permeability, and to monitor the growth of bacterial populations [21]. This kit comprises two fluorescent nucleic acid stains: SYTO9 and propidium iodide. SYTO9 (excitation and emission maxima, 480 and 500 nm) penetrates viable and nonviable bacteria, whereas propidium iodide (excitation and emission maxima, 490 and 635 nm) only enters bacteria with damaged plasma membranes [22,23], quenching the fluorescence of SYTO9. Thus, bacterial cells with compromised membranes fluoresce red, and those with intact membranes fluoresce green. Thin slices of pellets taken after 5 months of operation were analyzed by epifluorescence microscopy. Stained cells in the pellets were clearly observed in Figure 10a,b. Most cells in the pellets were intact, as evidenced by their green fluorescence Figure 10a, and several cells were dead, based on the PI labeling Figure 10b. The images of the living and dead cell were overlapped in Figure 10c. The dominant green color in the images implies that the pellet matrix supports the sufficient permeability of the substrate and oxygen, to promote microbial viabilities throughout the entire pellet.

## 4. Conclusions

In this paper, the immobilization method of the enriched microorganism was investigated for degrading TMAH and TOC in low-strength wastewater. The pellet without microorganism was prepared in various polymerization conditions and the TOC leak test was performed. The unstable pellets showed a breakthrough of TOC, while the stable pellet did not for a year. The PEG oligomer and 1.6% of crosslinker were found to be used for a highly cross-linked structure of stable pellet. DSC analysis results indicated a significant heat of curing from acrylate group in unstable pellets and almost negligible heat (<10 J/g) in stable pellets, due to unreacted acrylate block. This analysis can be used for sorting out polymerization conditions and controlling the quality of pellets. The inoculation method (biomass mixing ratio and total biomass concentration) and the operation conditions (HRT, packing density, and recirculation ratio) were optimized in terms of removal efficiencies in TMAH and TOC. Furthermore, the survival of the entrapped microorganism was confirmed by the viability test result. The immobilization technique has a great potential in the application of low-strength electronics wastewater treatment for reuse.

## Figures and Tables

**Figure 1 ijerph-17-04411-f001:**
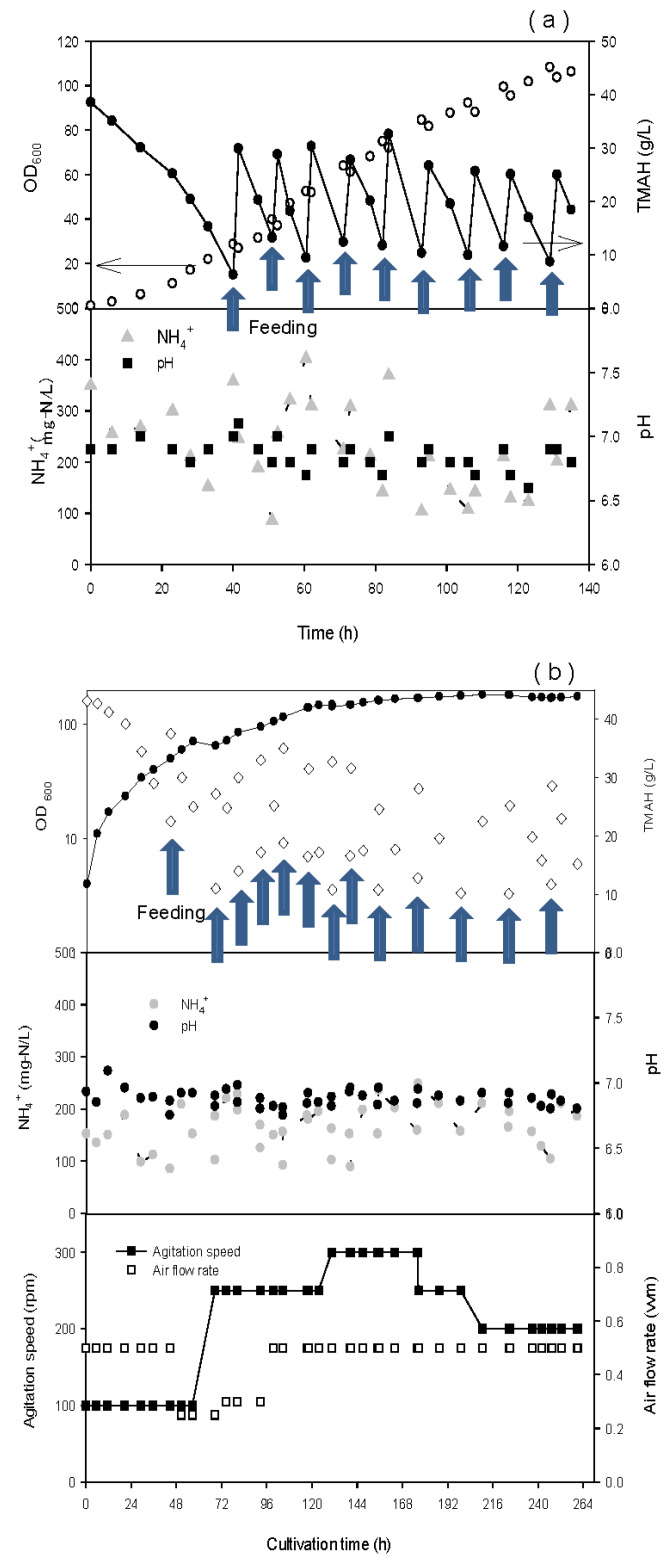
Cultivation of tetramethyl ammonium hydroxide (TMAH) degrading bacteria in (**a**) small scale (2.5 L) and (**b**) large scale (200 L).

**Figure 2 ijerph-17-04411-f002:**
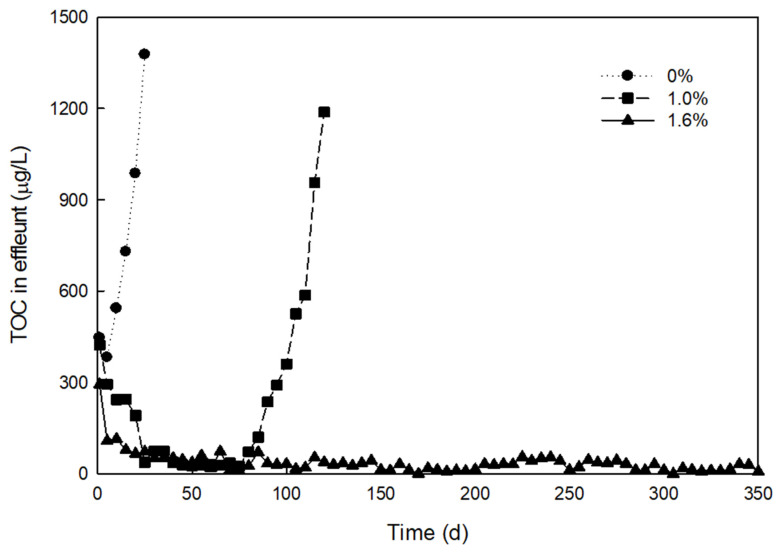
Concentration of TOC in eluents extracted from pellets prepared with different concentrations of crosslinker.

**Figure 3 ijerph-17-04411-f003:**
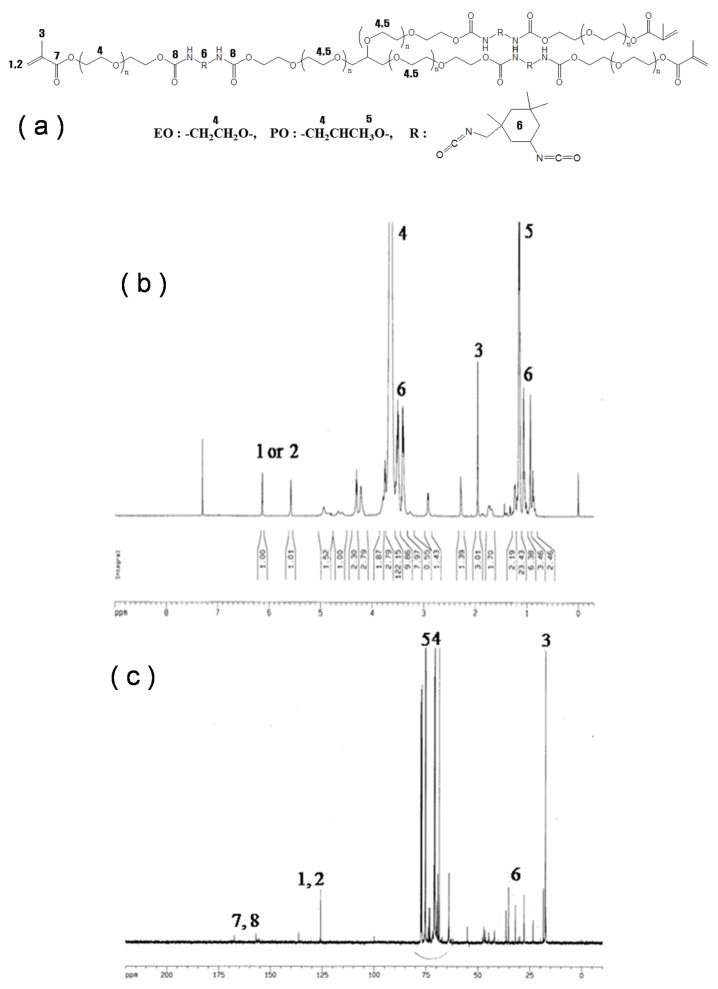
(**a**) Structure of oligomer and its spectra of (**b**) H-NMR and (**c**) C-NMR.

**Figure 4 ijerph-17-04411-f004:**
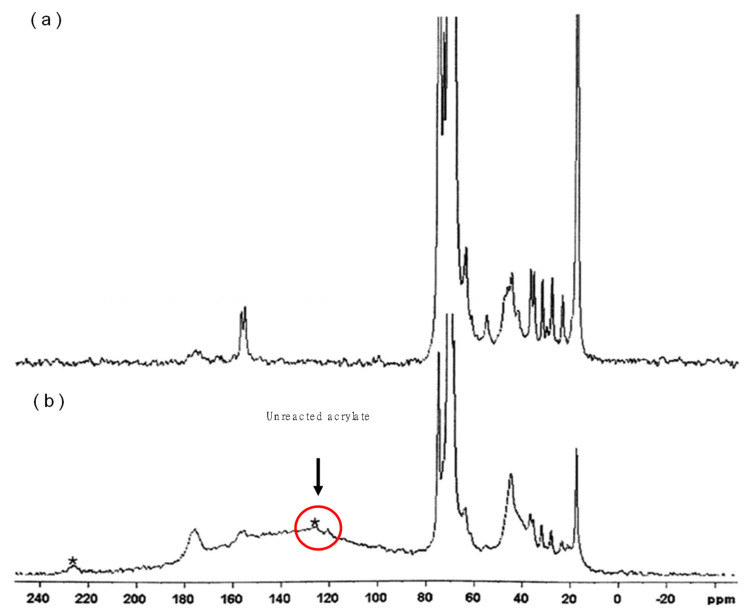
Solid NMR results of pellet prepared (**a**) with crosslinker (1.6%) (**b**) and without crosslinker (0%).

**Figure 5 ijerph-17-04411-f005:**
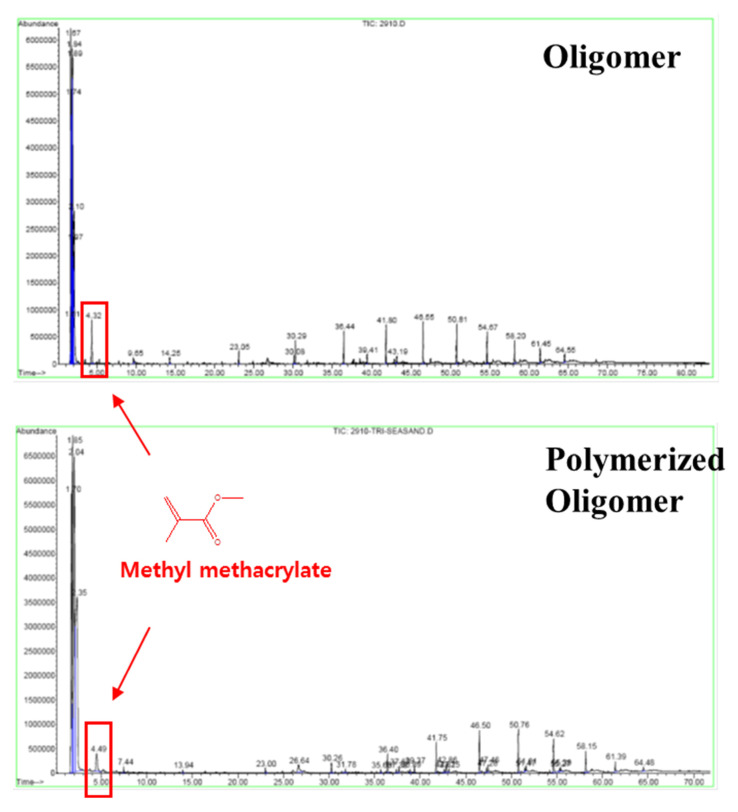
Pyrolysis gas chromatography (GC) chromatogram of oligomer and polymerized oligomer with spectrum representing methyl methacrylate.

**Figure 6 ijerph-17-04411-f006:**
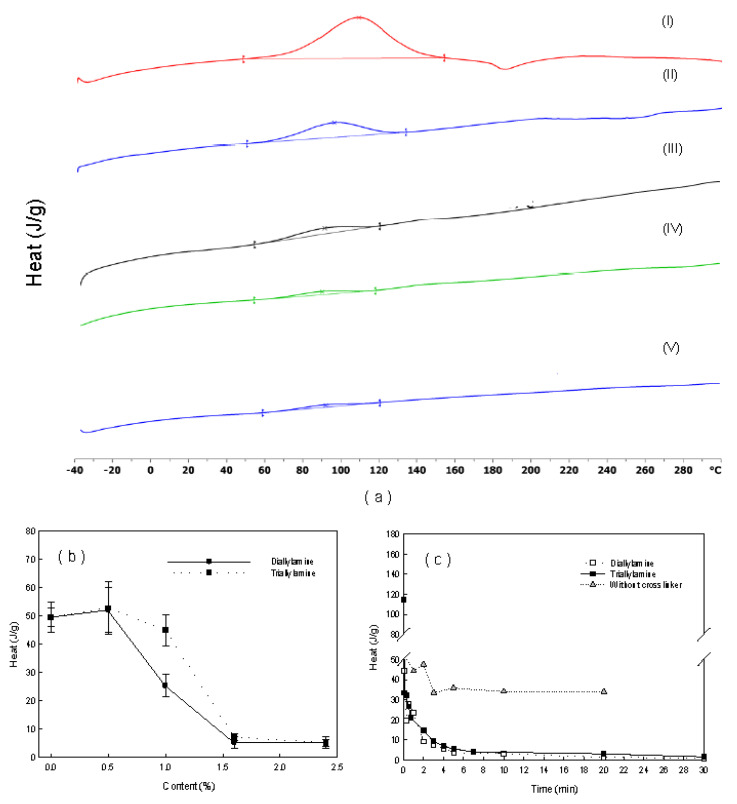
(**a**) DSC thermograms of pellets prepared with different concentrations of crosslinker (I: 0%, II:0.5%, III: 1.0%, IV: 1.6%, V: 2.4%), (**b**) area of curing peaks relative to amount of different concentrations of crosslinker with chemical initiator, and (**c**) area of curing peaks relative to UV exposure time with photoinitiator.

**Figure 7 ijerph-17-04411-f007:**
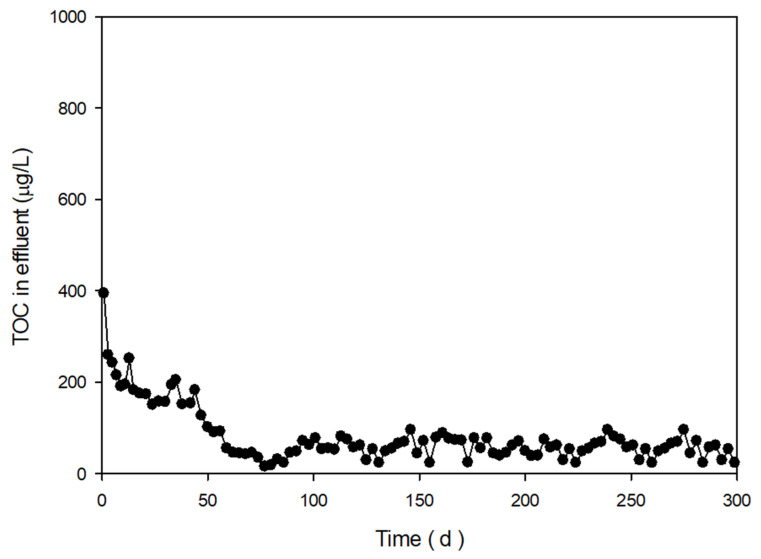
TOC leak test results of optimized pellets.

**Figure 8 ijerph-17-04411-f008:**
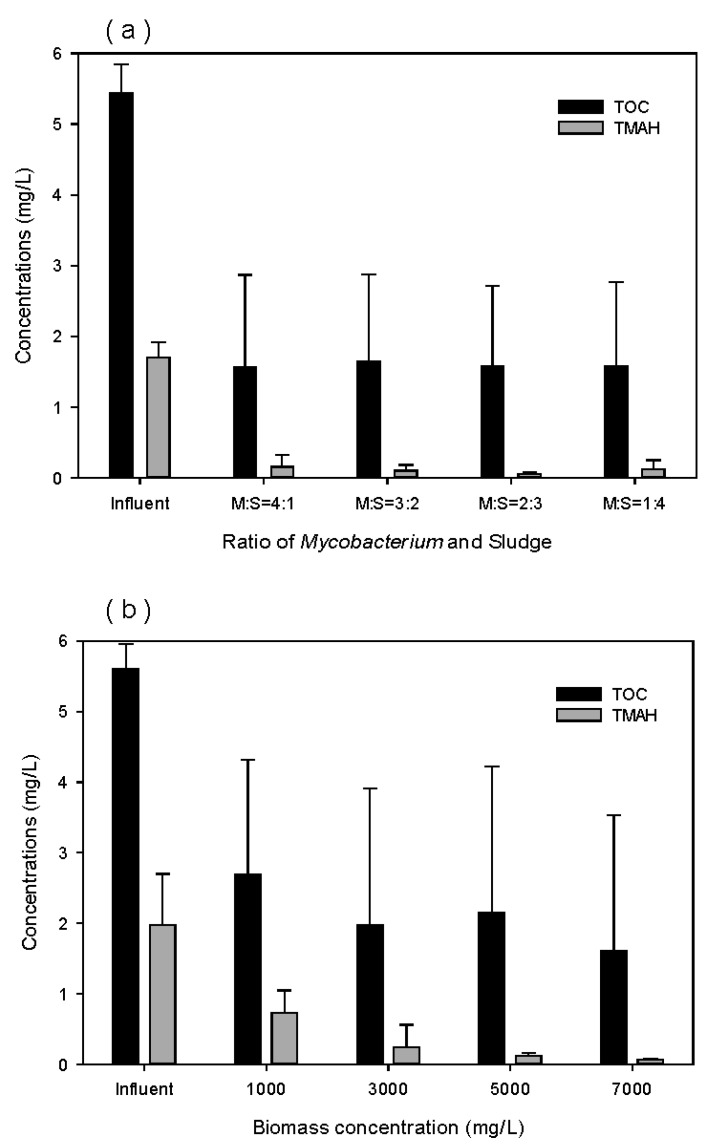
Removal efficiencies of TMAH with different (**a**) ratio of *Mycobacterium* sp. to sludge, and (**b**) biomass concentration.

**Figure 9 ijerph-17-04411-f009:**
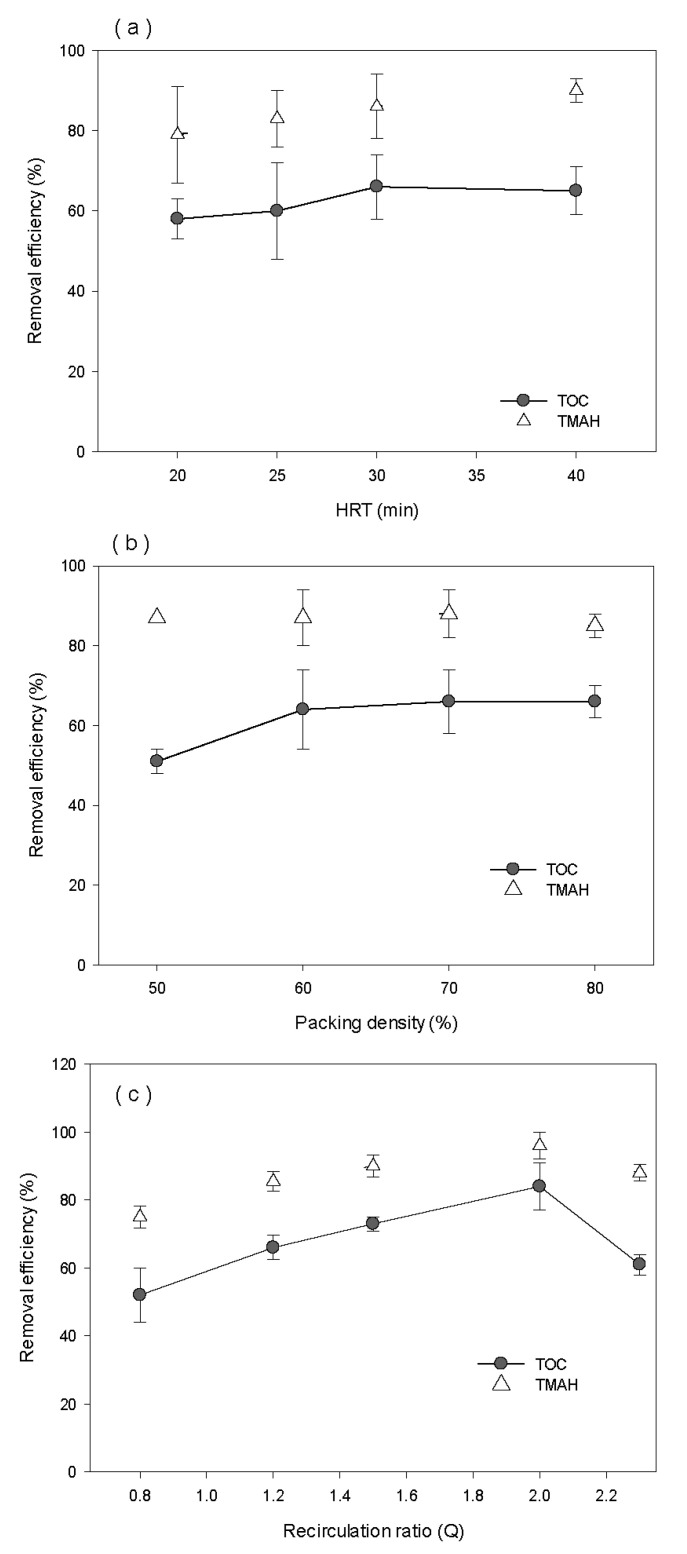
Removal efficiencies of TOC and TMAH in different (**a**) hydraulic retention time, (**b**) packing density, and (**c**) recirculation ratio.

**Figure 10 ijerph-17-04411-f010:**
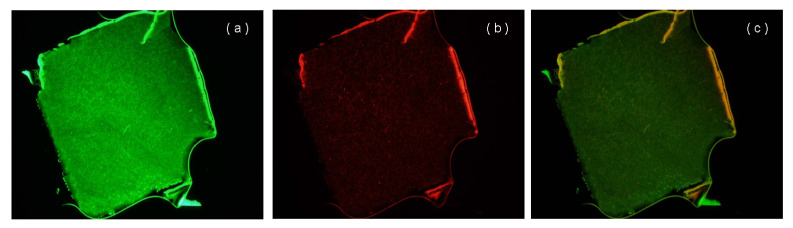
Viability of microorganism in pellets (**a**) for living cell only (**b**) for dead cell only (**c**) for living cell + dead cell.

**Table 1 ijerph-17-04411-t001:** Partial list of total organic carbon (TOC) leak test results in different preparation conditions.

No.	Type of Prepolymer	Type of Crosslinker	Dose of Crosslinker (wt%)	Dose of Initiator (mg/L)	^a^ Breakthrough Time (days)
1	Polyethylene glycol (PEG) monomer	^b^ DAA	1.0	10	7
2	PEG monomer	DAA	1.6	10	13
3	PEG monomer	DAA	2.4	10	28
4	PEG monomer	^c^ TAA	1.0	10	9
5	PEG monomer	TAA	1.6	10	19
6	PEG monomer	TAA	2.4	10	37
7	PEG oligomer	DAA	1.0	10	50
8	PEG oligomer	DAA	1.6	10	>360
9	PEG oligomer	DAA	2.4	10	>360
10	PEG oligomer	TAA	1.0	10	70
11	PEG oligomer	TAA	1.6	10	>360
12	PEG oligomer	TAA	2.4	10	>360

^a^: Breakthrough time was evaluated by time reached 100 μg/L of TOC in effluent; ^b^: DAA: diallyamine, ^c^: TAA: triallylamine.

**Table 2 ijerph-17-04411-t002:** Characteristics of low-strength electronics wastewater.

**pH**	4.84	**TOC**	4.392 mg/L
**Turbidity**	0.76 NTU	**TMAH**	1.998 mg/L
**T-N**	0.82 mg/L	**IPA**	1.620 mg/L
**T-P**	0.85 mg/L	**Ethanol**	0.026 mg/L
**NH_4_^+^-N**	0.05 mg/L	**Methanol**	0.590 mg/L
**COD**	10.58 mg/L	**Acetone**	0.022 mg/L

COD: chemical oxygen demand.

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
