# Peer review of "Development of Novel Method for Immobilizing TMAH-Degrading Microbe into Pellet and Characterization Tool, for Verifying Its Robustness in Electronics Wastewater Treatment"

_ijerph, 2020, doi:10.3390/ijerph17124411_

Round 1
Reviewer 1 Report
there are several studies on the immobilization of enzymes in polyethylene glycol, to remove various contaminants. A CRITICAL REVIEW of these articles MUST be made.
insert before TOC, total organic carbon
A discussion of the results with similar ones should be further developed (such as the paper "Low-strength electronic wastewater treatment using immobilized cells of TMAH-degrading bacterium followed by activated carbon adsorption" - https://doi.org/10.1080/19443994.2014.923192)
The methodology regarding the tests with the real effluent is very vague, with several gaps. the system where the experiment was developed must be detailed, with equipment, dimensions, etc.
It should be made explicit that the study is not with real effluent, but with THM solution. The real effluent may contain other contaminants, even if they were not detected by traditional characterization, which can interfere with the process.
The methods used in the characterization must be more explicit and referenced.
A statistical evaluation of the results is necessary.
THM removal results MUST be compared with similar jobs and other processes A further characterization of the bacterial films is necessary, as there may be, in addition to degradation, adsorption. Therefore the two processes must be quantified
Author Response
- There are several studies on the immobilization of enzymes in polyethylene glycol, to remove various contaminants. A CRITICAL REVIEW of these articles MUST be made.
Insert before TOC, total organic carbon
The following sentences were included in introduction part as follows,
“Among the immobilization method, it is reported that PEG-based method had good mechanical strength [12,13]. Most of these researches were focused on the application in nitrification and various organic compounds in municipal wastewater [14,15]. In low-strength electronics wastewater, contamination of TOC from microorganism and its support would be comparable to the concentration of TOC in effluent from wastewater treatment facility for reuse. However, no researches have been performed on the stability of prepared polymer despite its importance in low-strength wastewater.”
‘total organic carbon (TOC)’ was included in the text.
- A discussion of the results with similar ones should be further developed (such as the paper "Low-strength electronic wastewater treatment using immobilized cells of TMAH-degrading bacterium followed by activated carbon adsorption" - https://doi.org/10.1080/19443994.2014.923192)
We involved as a co-author in the mentioned article. This manuscript was focused on the microbial kinetics with immobilized pellet rather than polymerization stability. According to referee’s comment, we added the related content to introduction session.
“Also, the enrichment method of TMAH degrading bacteria and fundamental microbial kinetics were investigated [16].”
- The methodology regarding the tests with the real effluent is very vague, with several gaps. the system where the experiment was developed must be detailed, with equipment, dimensions, etc.
Lab column used for leak test with distilled water was used for real wastewater. The details in experimental apparatus for leak test was provided in materials and methods (2.3) as following,
“The stability of pellets without microorganism was evaluated by measuring TOC extracted from prepared pellets in a continuous mode. Pellets were filled in a glass column (4.0 cm ID x 20 cm height, 250 mL) and distilled water was supplied with 8.0 mL/min (HRT = 30 min) in upflow mode.”
- It should be made explicit that the study is not with real effluent, but with THM solution. The real effluent may contain other contaminants, even if they were not detected by traditional characterization, which can interfere with the process.
Real wastewater from electronics manufacturer was collected and used for tests under different operating conditions and long-term stability described in section 2.4. TMAH solution was used for enriching TMAH degrading microorganism for immobilization into pellet.
- The methods used in the characterization must be more explicit and referenced.
A statistical evaluation of the results is necessary.
As shown in figure 6, heat of curing below 10 J/g was required for good stability of immobilized pellet. This relationship was statistically significant (with over 10 samples).
- THM removal results MUST be compared with similar jobs and other processes A further characterization of the bacterial films is necessary, as there may be, in addition to degradation, adsorption. Therefore the two processes must be quantified
This paper focuses on the development of a characterization method using DSC chromatogram for replacing time-consuming stability test with distilled water and the verification under real wastewater with optimized pellets by developed characterization method. The comparison with other process (biological activated carbon) with developed pellets in this study is included in author’s another paper for pilot study. (J. of Haz. Mat. 384, 2020)
Reviewer 2 Report
The manuscript presents very interesting research results on an immobilization method of enriched microorganism for firmly degrading organic compounds including tetramethyl ammonium hydroxide in electronics wastewater without increase of TOC in effluent. The TMAH degrading bacteria was entrapped inside the pellets through polymerization.
For polymerization were used aqueous solution of polyethylene glycol (PEG) containing, PEG prepolymer and diallylamine or triallylamine as a crosslinker. The full characteristics of both the polymerization mixture and the final product, by using a differ method, are noteworthy. Polymerization conditions were optimized in terms of long-term TOC leak tests of pellet.
It has been shown, that the removal efficiency of TMAH was over 95% and effluent concentration of TOC was below 100 ppb. The viability test results revealed that entrapped microorganism were actively survived after 5 months of operations. The applied immobilization method can be considered as a new strategy for wastewater reuse process in low-strength electronics wastewater.
A very important element of the research was optimization work on the selection of crosslinking agent, which allowed the development of a procedure for the production of stable pellets.
Minor suggested corrections:
- Explanation of all abbreviations used, e.g. in the abstract -TOC, although all other abbreviations are explained.
- A more precise description of the subsection Enrichment of TMAH degrading bacteria, no information about the content of microorganisms. in the activated sludge from a wastewater plant.
- It is necessary to carefully check the manuscript to remove editorial errors (e.g. Section 2.4. Different size of font, Line 156 analySIS)
Author Response
The manuscript presents very interesting research results on an immobilization method of enriched microorganism for firmly degrading organic compounds including tetramethyl ammonium hydroxide in electronics wastewater without increase of TOC in effluent. The TMAH degrading bacteria was entrapped inside the pellets through polymerization. For polymerization were used aqueous solution of polyethylene glycol (PEG) containing, PEG prepolymer and diallylamine or triallylamine as a crosslinker. The full characteristics of both the polymerization mixture and the final product, by using a differ method, are noteworthy. Polymerization conditions were optimized in terms of long-term TOC leak tests of pellet. It has been shown, that the removal efficiency of TMAH was over 95% and effluent concentration of TOC was below 100 ppb. The viability test results revealed that entrapped microorganism were actively survived after 5 months of operations. The applied immobilization method can be considered as a new strategy for wastewater reuse process in low-strength electronics wastewater.A very important element of the research was optimization work on the selection of crosslinking agent, which allowed the development of a procedure for the production of stable pellets.
Minor suggested corrections:
- Explanation of all abbreviations used, e.g. in the abstract -TOC, although all other abbreviations are explained.
The following abbreviations are explained in the manuscript
‘total organic carbon (TOC)’ was included in the text.
‘(hydraulic retention time, HRT = 2 h)’
- A more precise description of the subsection Enrichment of TMAH degrading bacteria, no information about the content of microorganisms. in the activated sludge from a wastewater plant.
After completion of enrichment, the contents of TMAH-degrading bacteria increased (almost negligible at the beginning) up to 1.5%.
- It is necessary to carefully check the manuscript to remove editorial errors (e.g. Section 2.4. Different size of font, Line 156 analySIS)
Font size was corrected.
‘analySIS’ is the name of software.
Reviewer 3 Report
International Journal of
Environmental Research and Public Health
Manuscript number: ijerph-826212
Manuscript Title:” Development of Novel Method for Immobilizing TMAH-Degrading Microbe into Pellet and Characterization Tool for Verifying Its Robustness in Electronics Wastewater Treatment”
Recommendation: Accept after minor revision
Additional comments: The target of the manuscript is development of the effective method of immobilization of microorganism for robustly degrading organic compounds in wastewater coming from electronic industry. The peculiarity of this application is that water obtained in the results of this treatment should be then prepared as extremely high pure water.
This target is important and actual. So, the manuscript is dedicated to an actual problem of chemistry, of microbiology and of water purification technologies.
However, there are some problems in the manuscript that can be corrected.
It is not clear how the microorganisms are immobilized in polymer matrix to be stable during long term of use. So some results of studies by SEM or by TEM technique can be considered as essential for further understanding.
Author Response
Additional comments: The target of the manuscript is development of the effective method of immobilization of microorganism for robustly degrading organic compounds in wastewater coming from electronic industry. The peculiarity of this application is that water obtained in the results of this treatment should be then prepared as extremely high pure water.
This target is important and actual. So, the manuscript is dedicated to an actual problem of chemistry, of microbiology and of water purification technologies.
However, there are some problems in the manuscript that can be corrected.
- It is not clear how the microorganisms are immobilized in polymer matrix to be stable during long term of use. So some results of studies by SEM or by TEM technique can be considered as essential for further understanding.
We analyzed the image of immobilized microorganisms and published in another paper (attached file)
(Figure 3 in Analytical Sciences, 2008, 24, 547).

Round 2
Reviewer 1 Report
as previously mentioned, discussions of the results could be further developed. The statistical analysis of the results should have been carried out in ALL the tests, mainly in the adsorption tests.
However, the authors promoted a significant improvement in the body of the text.
Author Response
- As previously mentioned, discussions of the results could be further developed. The statistical analysis of the results should have been carried out in ALL the tests, mainly in the adsorption tests.
We added the description of statistical analysis results to the text:
“According to AVOVA test as statistical analysis, the relationship of pellet stability (TOC leakage in effluent less than 50 μg/L over 1 yr) to DSC analysis result (area of curing curve <10 J/g) was statistically significant (p<0.05), while that to other analysis results were not (p>0.05). Therefore, we determined this method as a characterization method for quality control of prepared pellets. This method is time-saving method compared with TOC leaking test of 1 year.”